

# Light intensity drives different growth strategies in two duckweed species: *Lemna minor* L. and *Spirodela polyrhiza* (L.) Schleiden

Małgorzata Strzałek and Lech Kufel

Institute of Biological Sciences, Faculty of Sciences, Siedlce University of Natural Sciences and Humanities, Siedlce, Poland

## ABSTRACT

Duckweed species *Lemna minor* and *Spirodela polyrhiza* are clonal plants with vegetative organs reduced to a frond and a root in *L. minor* or a frond and several roots in *S. polyrhiza*. They reproduce vegetatively by relatively rapid multiplication of their fronds. The habit of *S. polyrhiza* (large fronds with up to 21 roots) makes it a strong competitor among representatives of the family Lemnaceae, probably due to different resource-use strategies compared to small duckweed. In our study, light was the resource that affected the plants before and during the laboratory experiment. We sampled the plants from natural habitats differing in light conditions (open and shady) and grew them for 16 days in a thermostatic growth room at 22 °C under a 16:8 photoperiod and three light intensities (125, 236, 459 $\mu$mol photons m$^{-2}$ s$^{-1}$) to investigate the trade-off between frond enlargement and multiplication. Both species from the open habitat had higher growth rates based on the frond numbers and on surface area of fronds compared to plants from the shady habitat. They adopted different species-specific strategies in response to the experimental light conditions. The species size affected the growth rates in *L. minor* and *S. polyrhiza*. *Spirodela polyrhiza* grew slower than *L. minor*, but both species grew fastest at medium light intensity (236 $\mu$mol m$^{-2}$ s$^{-1}$). *Lemna minor* maintained the growth rates at high light intensity, while *S. polyrhiza* slowed down. *Spirodela polyrhiza* responded to deteriorating light conditions by increasing its frond surface area, thus optimising light capture. *Lemna minor* from the shady habitat enhanced light harvest by increasing chlorophyll *a* concentration, but did not invest more in frond enlargement than *L. minor* from the open habitat. Under shady conditions, *S. polyrhiza* is likely to achieve an advantage over *L. minor* due to the larger frond size of the former. Our findings suggest the existence of a trade-off between size and number in duckweed.

## INTRODUCTION

*Lemna minor* L. and *Spirodela polyrhiza* (L.) Schleiden, belonging to the Lemnaceae family, are widespread clonal species known as duckweed (*Landolt, 1986*). They form free-floating mats in open and shady habitats of stagnant and slow-flowing freshwater, especially those

Corresponding author
Lech Kufel, lech.kufel@uph.edu.pl

rich in nutrients (*Landolt, 1986*; *Kufel et al., 2012*). Their vegetative organs are reduced to a flat leaf-like frond with one root in *L. minor* or many roots (up to 21) in *S. polyrhiza*. The latter has fronds twice as large as those of *L. minor* and also the largest in the family (Ø 4–12 mm). The size of the fronds and even their shape depend on the external conditions (i.a. light intensity, photoperiod, nutrient concentrations, temperature) and genetic variability among clones (*Landolt, 1986*).

Both duckweed species reproduce mainly vegetatively by meristems inside lateral pouches. Daughter fronds grow alternately from two mother pouches and remain attached to the mother frond *via* stipes for some time, forming a colony. This method of propagation allows the duckweed to quickly colonise free spaces (*Landolt, 1986*; *Lemon & Posluszny, 2000*; *Ziegler et al., 2015*; *Acosta et al., 2021*). As clonal plants, they avoid the trade-off between longevity and the rate of offspring production, but they may potentially sacrifice the size of fronds for their numbers (*Stuefer, Van Hulzen & During, 2002*; *Aarssen, 2008*). According to the life history theory, the large allocation of resources in size precludes a high reproduction rate and vice versa (*Stearns, 1992*). Any plant strategy is affected by environmental conditions (*Stuefer, Van Hulzen & During, 2002*). Therefore, one may expect that in open and disturbed habitats, a genet should succeed by producing small but numerous daughter fronds, as opposed to undisturbed shady habitats where the production of large fronds may be more beneficial. The rationale for these suppositions is provided by the 'physical-space-niche size distribution' hypothesis that small individuals use space and associated patchy resources more efficiently than large individuals, while large plants provide higher fecundity (frond production) and a better ability to compete (*Aarssen, Schamp & Pither, 2006*).

Light is a key factor affecting the growth and reproduction of plants. The response of duckweed to light availability depends on species, clone, ambient temperature and tissue nutrient supply. Different light-dependent metabolic processes have different light requirements, which may additionally be affected by co-occurring stressors and biotic factors, *e.g.*, competition (*Landolt & Kandeler, 1987*; *Valladares & Niinemets, 2008*). In general, light saturation for growth rates is lower than for photosynthetic rates and ranges for different Lemnaceae species and clones from 5,000 to 15,000 lux (*Landolt & Kandeler, 1987*), corresponding to a photosynthetic photon flux density (PPFD) of 68–203 $\mu$mol photons m$^{-2}$ s$^{-1}$ of cool white light (*Thimijan & Heins, 1983*). At the same temperature, *S. polyrhiza* reached the maximum growth rate of 0.38 day$^{-1}$ at about 150 $\mu$mol m$^{-2}$ s$^{-1}$, whereas *L. minor* needed more than twice as much light to reach the plateau of the growth curve at 0.41 day$^{-1}$ (*Landolt & Kandeler, 1987*). In *L. minor*, high light intensity changes the composition of leaf pigments and increases plant biomass, frond size, root length, the content of proteins and starch (*Landolt & Kandeler, 1987*; *Lepeduš et al., 2020*; *Stewart et al., 2021*). Chlorophyll *a* is the main photosynthetic pigment that, together with chlorophyll *b* and carotenoids, play an essential role in capturing light energy, and their ratios in leaves vary with light availability (*Lepeduš et al., 2020*; *Stewart et al., 2021*). Low PPFD is accompanied by a decrease in the chlorophyll *a:b* ratio and an increases in total chlorophyll concentration (*Paolacci, Harrison & Jansen, 2018*; *Lepeduš et al., 2020*; *Stewart et al., 2021*). Based on the composition of leaf pigments, *Stewart et al. (2021)* clustered

duckweed species (*Lemna gibba* L. and *L. minor*) with heliophilous perennials and highly shade-tolerant evergreens, indicating their ability to grow under varying light conditions.

During our studies carried out in *Stratiotes aloides* L. stands (*Kufel et al., 2010*; *Kufel et al., 2012*; *Strzałek, Kufel & Wysokińska, 2019*), we observed the dominance of various Lemnaceae species in mixed-species communities growing among *Stratiotes* leaves, therefore we decided to conduct a series of experiments explaining the reasons for our field observations. This experiment tested how basic life history traits, such as growth rates measured by the surface area and the number of fronds in two duckweed species most commonly occurring in *Stratiotes* stands, *L. minor* and *S. polyrhiza*, are affected by light intensity prevailing both in their habitats and under experimental conditions. These traits may affect the outcome of interspecific competition in duckweed communities. The larger species further limit the size of the smaller species by reducing light and nutrients, and the smaller species constrain the larger species by reducing space for offspring (*Aarssen, 2008*). We hypothesised that *L. minor* as a smaller species multiplicates faster than *S. polyrhiza* and *S. polyrhiza* as a larger species invests more in enlarging its own fronds. The species respond differently to experimental light intensities, and the effect depends on light conditions in their natural habitats. Since Lemnaceae are used in wastewater phytoremediation, biofuel production and animal and human nutrition (*e.g.*, *Cheng & Stomp, 2009*; *Gupta & Prakash, 2013*; *Appenroth et al., 2017*; *Liu et al., 2019*; *Acosta et al., 2021*), our results may have more wide-ranging practical applications.

## MATERIALS & METHODS

### Sampled habitats

Plants for the experiment were collected in September 2019 from two locations—a small (0.06 ha) and shallow pond in the village of Miednik (52°31′34″N; 21°57′36″E) and an embayment of the Liwiec River near the town of Węgrów (52°23′51″N; 22°00′20″E) in central-eastern Poland. Throughout the growing season, the water surface in the pond is shaded by the surrounding deciduous trees (mainly *Acer* spp. and *Quercus robur* L.) and shrubs shedding leaves for winter and by evergreen pines *Pinus sylvestris* L. (shady habitat), while that in the embayment is open (open habitat). Light intensity on a sunny sampling day was 2,374 µmol photons m$^{-2}$ s$^{-1}$ on the surface of the open water body and 121 µmol photons m$^{-2}$ s$^{-1}$ on the shaded pond, while at the same time in the nearby open area it was 1,980 µmol photons m$^{-2}$ s$^{-1}$. *Lemna minor* and *S. polyrhiza* co-occurred at both sites.

### Experiment

Bulk plant material was brought to the laboratory, washed with tap water, placed in containers (separately for each habitat) filled with synthetic N medium (*Appenroth, Teller & Horn, 1996*) and left for a four-day preculture period (*Appenroth, 2002*). The plants divided into the species were then transferred to perforated transparent PET cups (80 mL, Ø 70 mm) immersed in cuvettes filled with 2 L of nutrient medium. Each cuvette contained ten cups with three colonies (9–13 fronds per cup) of *L. minor* and ten cups with two colonies (4–10 fronds per cup) of *S. polyrhiza* from a given habitat (shady or open) randomly distributed in the respective cuvette. In total, the number of cups was

as follows: 10 replicates × 2 species × 2 habitats × 3 light intensities = 120. Three light intensities were applied with the following average PPFD values: low – 125, medium – 236 and high – 459 $\mu$mol photons m$^{-2}$ s$^{-1}$ at a light:dark ratio of 16:8 h. The value of high light intensity was based on light saturation points for duckweed growth given in *Landolt & Kandeler (1987)*. Light was provided by cool-white fluorescent tubes and its intensity was measured using a LI-192 sensor and a LI-250A light meter (LI-COR Inc., USA). The experiment was carried out in a thermostatic growth room at a temperature of 22 °C and lasted 16 days. The nutrient medium was replaced in each cuvette every four days to inhibit algal development and to prevent nutrient depletion. The number of colonies and fronds in each cup was counted on the same days and the positions of the cups within the cuvettes were randomised. On days 1 and 16, fronds from each cup were photographed using a Sony $\alpha$500 camera equipped with macro lens DT 2.8/30. The obtained photos were processed using the Corel Draw Photo Paint X5 package and then used to determine the surface area of all fronds using ImageJ software.

Plant performance was measured with two parameters. The first was the growth rate based on the number of fronds (rN) calculated according to the following equation describing exponential growth:

$$rN = (\ln N_t - \ln N_0)/t,$$

where $N_0$ and $N_t$ represent the number of fronds at the beginning and at the end of the experiment, respectively, and $t = 16$ days. The second parameter, the growth rate based on the surface area of fronds (rS), was calculated using the following formula:

$$rS = (\ln S_t - \ln S_0)/t,$$

where $S_t$ and $S_0$ are surface areas of fronds at the end and at the beginning of the experiment, respectively, and $t = 16$ days. The exponential model rN was checked using data from measurements performed every four days from the beginning to the end of the experiment (five occasions).

Chlorophyll *a* was extracted in 90% acetone from fresh plant weight (FW) at the beginning and at the end of the experiment in three replicates ($n = 48$). The extracts were measured spectrophotometrically and translated into chlorophyll concentrations following the equations provided by *Golterman (1969)*. We adopted an extinction coefficient of $K = 89$ and replaced the filtrate volume by the FW sample in grams in the corresponding equations.

## Statistical analysis

MANOVA and univariate F tests were performed on the transformed data to test the effects of light intensities, habitats, plant species and factor interactions on the growth rates. The Box–Cox transformation normalised the data distribution and homogenised the variance. Three-way ANOVA (3 × 2 × 2) was applied to test differences between chlorophyll *a* concentrations in plant tissues. One-way ANOVA was used to test differences between the frond sizes in the plant species from different habitats at the beginning of the experiment. MANOVA and ANOVA were followed by Tukey multiple comparison test. Due to the lack of normality (Shapiro–Wilk test) and/or variance heterogeneity (Bartlett test), Kruskal–Wallis ANOVA and post-hoc Dunn test were used for differences between the growth rates
($\Delta r = rS - rN$), frond sizes and the frond size increment ($S_{end}/S_{start}$ = frond surface area at the end of the experiment/frond surface area at the beginning of the experiment). Lines of best fit were obtained using nonlinear and linear regressions where appropriate. To meet the assumptions of linear regression, two outliers were removed from the analysis. A significance level of 0.05 was assumed. All statistical analyses were conducted using *Statistica* 13.3 (TIBCO Software Inc.). The chlorophyll *a* concentrations presented in the Results section are mean values and their standard deviations.

## RESULTS

The number of fronds in all experimental light conditions increased with time, and the changes were well fitted to the exponential model—see examples presented in Fig. 1.

MANOVA identified three factors and two interactions as significant for the growth rates based on the surface area of fronds (rS) and the number of fronds (rN) in *L. minor* and *S. polyrhiza* (Table 1). Univariate F tests revealed that rS and rN for both species, irrespective of habitat, were highest at medium light intensity ($F_{2,108} = 14$, $p < 0.001$ and $F_{2,108} = 22$, $p < 0.001$ for rS and rN, respectively; Fig. 2). Plants from the open habitat performed better than the duckweed from the shade (rS: $F_{1,108} = 351$, $p < 0.001$; rN: $F_{1,108} = 157$, $p < 0.001$) and *S. polyrhiza* grew slower than *L. minor* (rS: $F_{1,108} = 84$, $p < 0.001$; rN: $F_{1,108} = 69$, $p < 0.001$). Although both duckweed species had their optima for rS and rN at about 236 $\mu$mol photons m$^{-2}$s$^{-1}$ (medium light intensity), they adopted different strategies in response to experimental light conditions (rS: $F_{2,108} = 3$, $p = 0.037$; rN: $F_{2,108} = 20$, $p < 0.001$). *Spirodela polyrhiza* fronds, irrespective of habitat, grew more slowly at high light intensity (Tukey test: $p < 0.001$, $df = 108$ for rS and $p < 0.001$, $df = 108$ for rN), while *L. minor* performed at the same rate at medium light intensity (Tukey test: $p = 0.37$, $df = 108$ for rS and $p = 0.73$, $df = 108$ for rN; Fig. 2). The interaction between the duckweed species and their habitats was significant only for rN ($F_{1,108} = 35$, $p < 0.001$). Both species from the shade multiplicated (rN) at the same rate (Tukey test: $p = 0.32$, $df = 108$) and slower than the plants from the open habitat (Tukey test: $p < 0.019$, $df = 108$). The fastest multiplication was observed for *L. minor* from the open habitat (Tukey test: $p < 0.001$, $df = 108$) and then for *S. polyrhiza* from the open habitat (Tukey test: $p < 0.001$, $df = 108$). The mean values of rN and rS and their standard deviations are presented in Table S1.

In both *L. minor* and *S. polyrhiza,* the growth rate based on the frond surface area (rS) was higher than that based on the number of fronds at all light intensities (Fig. 2). This means that the multiplication was accompanied by an enlargement of plant fronds. The difference $\Delta r = rS - rN$ was always positive and may be considered here as a relative investment of plants in increasing the size of fronds rather than their multiplication. In general, the investment of the species from the same habitat did not differ significantly with respect to light intensity (Table 2). The relation between $\Delta r$ and the habitat was observed only for *S. polyrhiza*. At the beginning of the experiment, fronds of *S. polyrhiza* from the shade were on average 1.3 times larger than those from the open habitat (one-way ANOVA, $F_{1,58} = 25.66$, $p < 0.001$). *Spirodela polyrhiza* from the open habitat responded with frond enlargement to the deterioration of light conditions in the culture relative to

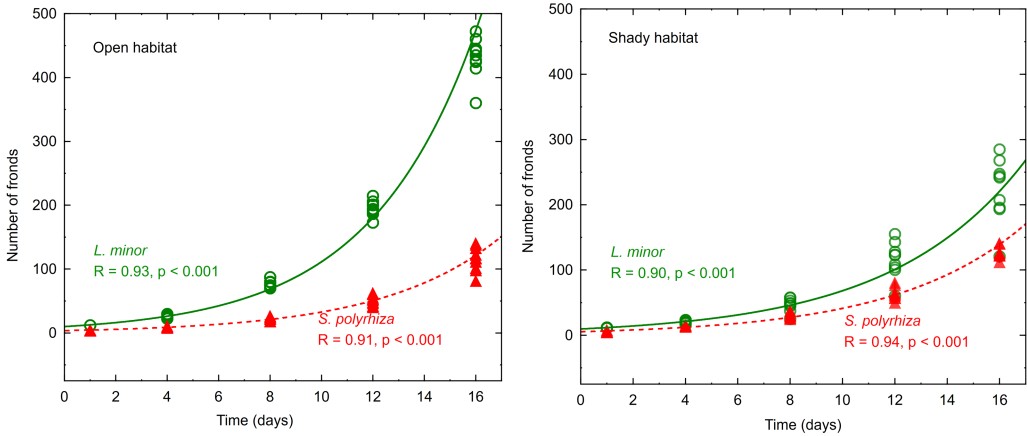

**Figure 1** **Increasing number of fronds during the experiment fitted to an exponential model.** Data represent plants from the open and shady habitats grown at low light intensity ($n = 10$).

**Table 1** **Effect of light intensity (low, medium, high), habitat (open or shady), species and their interactions on the growth rates of *L. minor* and *S. polyrhiza* (MANOVA).**

| Factors | F | Df effect | Df error | p |
|---|---|---|---|---|
| **Light intensity** | 10 | 4 | 214 | **<0.001** |
| **Habitat** | 178 | 2 | 107 | **<0.001** |
| **Species** | 44 | 2 | 107 | **<0.001** |
| Light intensity * habitat | 1 | 4 | 214 | 0.66 |
| **Light intensity * species** | 12 | 4 | 214 | **<0.001** |
| **Habitat * species** | 27 | 2 | 107 | **<0.001** |
| Light intensity * habitat * species | 1 | 4 | 214 | 0.47 |

**Notes.**

Light intensities: low–125, medium–236 and high –459 $\mu$mol photons m$^{-2}$ s$^{-1}$. Statistically significant effects are indicated in bold.

the natural habitat and doubled the mean frond size by the end of the experiment (Table 2; Fig. 3). Interspecific differences in the investment were statistically significant only within the shady habitat and low light intensity: *L. minor* from the shade invested more in its growth than co-occurring *S. polyrhiza* (Dunn test: $p = 0.005$, $n = 120$).

The regression of the frond size increment on rN was significant for both duckweed species regardless of the habitat they came from (Fig. 4). This relationship was least pronounced in *L. minor* from the shade ($\beta = -0.37$) and most pronounced in co-occurring *S. polyrhiza* ($\beta = -0.61$). In the case of duckweed from the open habitat, $\beta$ had similar values ($-0.48$ for *L. minor* and $-0.40$ for *S. polyrhiza*). The regression analysis showed that the multiplication of fronds reduced their enlargement mainly in *L. minor* from the open habitat. *Spirodela polyrhiza* from the shade had sufficiently large fronds at the beginning of the experiment that it did not need to enlarge them further and could allocate resources to multiplication. The relatively low coefficients of determination were due to considerable variation in the size and/or number of fronds within low and high light intensities (CV = 10–16%).

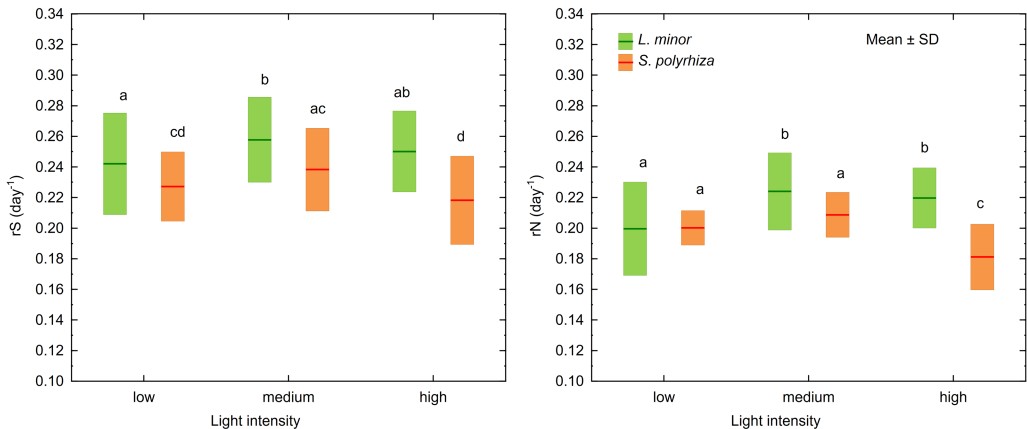

**Figure 2  Growth rates as determined by the surface area of fronds (rS) and the number of fronds (rN) in *L. minor* and *S. polyrhiza* at different light intensities.** Each box combines data from both habitats. Different letters denote statistically significant differences in Tukey test ($p < 0.05$, $n = 20$) following MANOVA.

**Table 2  Medians of the difference ($\Delta$r) between the growth rate based on the surface area of fronds (rS) and the number of fronds (rN) in *L. minor* and *S. polyrhiza* from the open and shady habitats grown at different light intensities.**

| Species | Light intensity | $\Delta$r | |
|---|---|---|---|
| | | **Open** | **Shady** |
| | low | 0.043[ah] | 0.041[ab] |
| *L. minor* | medium | 0.035[aeh] | 0.035[abd] |
| | high | 0.037[afgh] | 0.024[bef] |
| | low | 0.040[afh] | 0.014[cde] |
| *S. polyrhiza* | medium | 0.044[ah] | 0.014[cdeg] |
| | high | 0.049[a] | 0.018[bch] |

**Notes.**
Light intensities: low– 125, medium– 236 and high – 459 $\mu$mol photons m$^{-2}$ s$^{-1}$. The significance of differences in the investment into the two growth parameters of both plants was checked by Dunn test following Kruskal–Wallis ANOVA ($H_{11,120} = 74.73$, $p < 0.001$) and marked with different letters ($p < 0.05$).

In addition, we analysed the initial and final concentrations of chlorophyll *a* in the plants studied. Three-way ANOVA showed a significant effect of light intensity, plant habitat, and the latter together with species on chlorophyll *a* concentrations (Table 3). The duckweed growing under low light intensity had the highest concentration of the pigment ($1.089 \pm 0.106$ mg g$^{-1}$ FW). The increase in light intensity caused a gradual decrease in the concentration of chlorophyll *a* in plant tissues ($0.904 \pm 0.118$ at medium light intensity, $0.816 \pm 0.101$ mg g$^{-1}$ FW at high light intensity) to the initial level recorded at the beginning of the experiment ($0.838 \pm 0.111$ mg g$^{-1}$ FW). Despite different light intensities used, plants from the shade contained on average more chlorophyll *a* ($0.955 \pm 0.145$ mg g$^{-1}$FW) than those from the open habitat ($0.869 \pm 0.149$ mg g$^{-1}$ FW). Habitat had no effect on the concentration of chlorophyll *a* in *S. polyrhiza*, while *L. minor* from the shady habitat had the highest chlorophyll *a* concentration of all plants studied (Fig. 5).

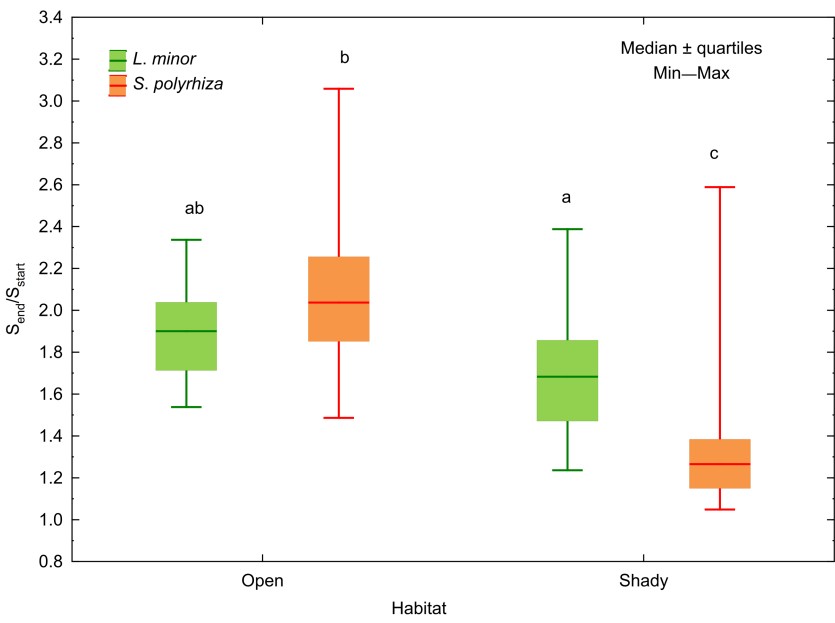

**Figure 3** Increase in the surface area of fronds between the start ($S_{start}$) and the end ($S_{end}$) of the experiment in relation to the habitat of *L. minor* and *S. polyrhiza*. Each box combines data from three light intensities. Different letters denote statistically significant differences between species from the two habitats in Dunn test ($p < 0.01$) following Kruskal–Wallis ANOVA ($H_{3,120} = 58.60$, $p < 0.001$).

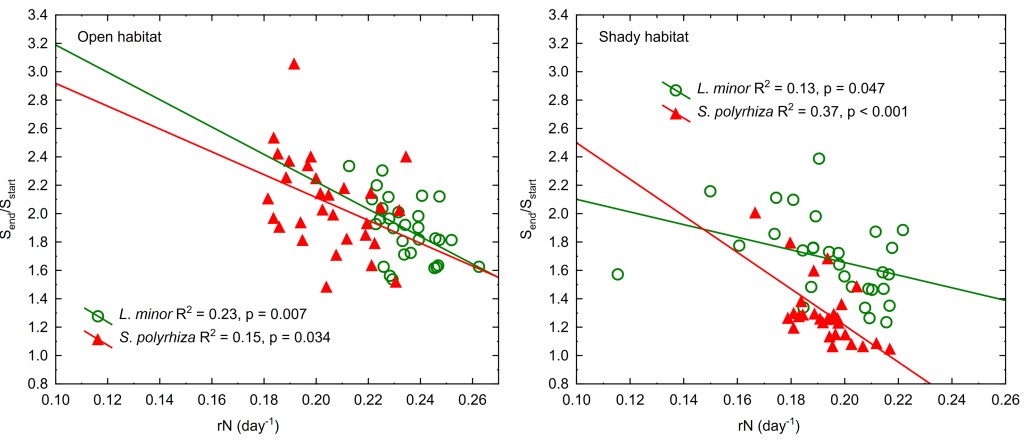

**Figure 4** Regression lines of frond size increments ($S_{end}/S_{start}$) on the growth rate based on the number of fronds in *L. minor* and *S. polyrhiza* from two habitats ($n = 30$). $S_{start}$ indicates the surface area of fronds at the beginning of the experiment, $S_{end}$ indicates the surface area of fronds at the end of the experiment.

## DISCUSSION

The response of Lemnaceae to light intensity depends on many factors, including species, clone, temperature, nutrient concentrations, $CO_2$ supply and light quality (*Landolt & Kandeler, 1987*). Both species, *L. minor* and *S. polyrhiza*, are small compared to other

**Table 3** Effect of light intensity (low, medium, high), habitat (open or shady), species (*L. minor*, *S. polyrhiza*) and their interactions on chlorophyll *a* concentrations in the studied plants (three-way ANOVA).

| Factors | SS | df | MS | F | p |
|---|---|---|---|---|---|
| **Light intensity** | 554434 | 3 | 184811 | 35.43 | **<0.001** |
| **Habitat** | 88696 | 1 | 88696 | 17.00 | **<0.001** |
| Species | 608 | 1 | 608 | 0.12 | 0.74 |
| Light intensity * habitat | 38097 | 3 | 12699 | 2.43 | 0.08 |
| Light intensity * species | 16601 | 3 | 5534 | 1.06 | 0.38 |
| **Habitat * species** | 200706 | 1 | 200706 | 38.48 | **<0.001** |
| Light intensity * habitat * species | 12523 | 3 | 4174 | 0.80 | 0.50 |
| Error | 166922 | 32 | 5216 | | |

**Notes.**

Light intensities: low– 125, medium– 236 and high – 459 $\mu$mol photons m$^{-2}$ s$^{-1}$. Statistically significant effects are indicated in bold.

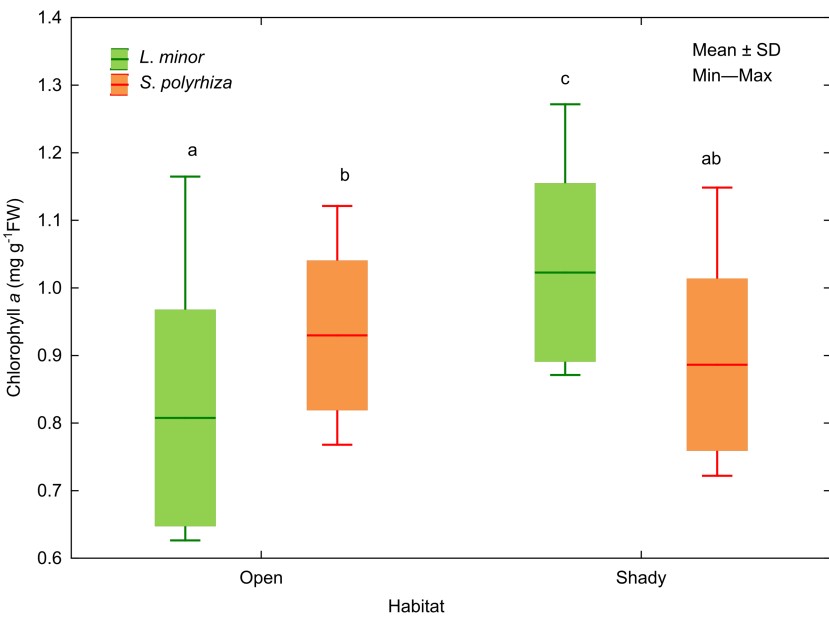

**Figure 5** Effect of habitat–species interaction on the concentration of chlorophyll *a* in *L. minor* and *S. polyrhiza*. Each box combines initial concentrations and experimental data from three light intensities. Different letters denote statistically significant differences in Tukey test ($p = 0.002$) following three-way ANOVA–see Table 3.

representatives of angiosperms, but even small differences in size can give a competitive advantage, especially under the conditions of competition for light, and generate different life strategies (*Aarssen, Schamp & Pither, 2006*). In our study, *S. polyrhiza*, being larger, grew slower than smaller *L. minor*, which was also noted by *Lemon, Posluszny & Husband (2001)*. Their study revealed that *S. polyrhiza* retained daughter fronds longer than *L. minor* and produced only a few new fronds during that time. Such long retention results in larger fronds and slower population growth. These results are consistent with competitive strategies of

larger species that take advantage of their size, while smaller species like *L. minor* gain a numerical advantage (*Aarssen, 2008*). The multiplication of fronds in duckweed can be regarded as equivalent to seed production in sexually propagating species. The latter is higher in small plants than in large ones (*Aarssen, Schamp & Pither, 2006*), thus it is not surprising that in our experiment *L. minor* had a higher rN than *S. polyrhiza*.

In our study, experimental light conditions affected the growth rates in both duckweed species, indicating species-specific light strategies. Both species grew fastest at medium light intensity (about 236 $\mu$mol photons m$^{-2}$ s$^{-1}$), which slightly exceeded the upper limit of light saturation range of 15,000 lux (about 203 $\mu$mol photons m$^{-2}$ s$^{-1}$) for duckweed growth under continuous light (*Landolt & Kandeler, 1987*). The higher light intensity resulted in a marked reduction in the growth rate of *S. polyrhiza* as determined by the number of fronds and a slightly smaller decrease in the growth rate based on the frond surface area, while *L. minor* maintained similar growth rates. The clear differences in light requirements of both species are consistent with the results presented by *Landolt & Kandeler (1987)* according to which *S. polyrhiza* achieved its optimal growth rate at about 250 $\mu$mol photons m$^{-2}$ s$^{-1}$, while *L. minor* needed about 100 $\mu$mol photons m$^{-2}$ s$^{-1}$ more to reach its optimum. In duckweed growth inhibition tests, 100 $\mu$mol photons m$^{-2}$ s$^{-1}$ of continuous light is recommended (*e.g.*, ISO 20079, 2005 cited in *Ziegler et al., 2015*), which corresponds to the low light intensity applied in our experiment. Our clones of both species performed better under the higher light level and a 16:8 photoperiod, but multiplicated more slowly than under axenic conditions. In the following discussion, we will avoid direct comparisons with data from the literature because, as argued by *Ziegler et al. (2015)*, they can only apply to the same clones cultivated under the same conditions.

We showed that light conditions in the natural habitat of duckweed species affect their growth rate as determined by the number of fronds, and thus our findings may confirm that this rate reflects the adaptation of clones to local conditions (however, see caveats raised by *Ziegler et al., 2015*). The adaptation may involve genetic modifications of light-dependent metabolic pathways in plants. However, understanding its mechanism was not a goal of our study. We found that both species from the open habitat multiplicated faster than duckweed from the shady habitat.

Our results showed different strategies of growth investments in *L. minor* and *S. polyrhiza* depending on habitat for *S. polyrhiza* and experimental light intensities for *L. minor*. *Spirodela polyrhiza* from the open habitat had considerably smaller fronds compared to plants from the shade and enlarged them during the experiment in response to a reduction of ambient light. This is also evidence for phenotypic plasticity of *S. polyrhiza* in adapting to local environmental conditions. The enlargement of the relatively large fronds appears to be sufficient for them to thrive and did not require an additional increase in chlorophyll *a* concentration in their tissues to optimise light capture. In contrast, *L. minor* from both habitats invested nearly equally in the growth of fronds at the same light intensity. Living in prolonged shade was associated with maintaining a rather high concentration of chlorophyll *a* in the tissues of this small species. Our findings contradict the conclusion of *Landolt & Kandeler (1987)* that high light intensities promote frond enlargement. It should be noted that plants from the shady habitat in our study adjusted to higher experimental light

intensities, while plants from the open habitat adjusted to low light intensities. The response of plants to shading involves a variety of mechanisms ranging from structural ones related to plant habit to molecular changes. Increasing the surface area of assimilatory organs (fronds) and remodelling the photosynthetic apparatus to capture sufficient light energy to thrive are among them (*Valladares & Niinemets, 2008*; *Lepeduš et al., 2020*; *Stewart et al., 2021*). Similar to the study by *Lepeduš et al. (2020)* on *L. minor*, exposure to higher light intensity during our experiment was associated with a decrease in chlorophyll *a* concentrations in both duckweed species.

Our results indicate that there is an inverse relationship between the multiplication rate and frond size increment in both *L. minor* and *S. polyrhiza*, but two of the four regression models were at the limit of statistical significance. This was due to intensive multiplication with simultaneous frond enlargement in *S. polyrhiza* from the open habitat and fairly slow multiplication and little frond enlargement in *L. minor* from the shady habitat. Clonality allows for economical propagation of infinitely long-lived genets without producing large adults and fertilising ovules (*Aarssen, 2008*).

Apart from Landolt's monographs (*Landolt, 1986*; *Landolt & Kandeler, 1987*), there are few papers comparing the traits of these two common duckweed species, *i.e., L. minor* and *S. polyrhiza* (*Bergmann et al., 2000*; *Lemon & Posluszny, 2000*; *Lemon, Posluszny & Husband, 2001*; *Kufel et al., 2012*; *Ziegler et al., 2015*; *Acosta et al., 2021*). Most of the studies cited herein concern clones cultivated under axenic conditions, which have never been in contact with their natural environment. In our research, we used plants freshly collected in the field to bring the experimental conditions closer to the real environment.

## CONCLUSIONS

Our results indicate that the species size affects the growth rates in *L. minor* and *S. polyrhiza*. As hypothesised, *L. minor* being a smaller species performed better than larger *S. polyrhiza*, and light conditions in their natural habitats and in the laboratory modified the studied traits. Plants from the open habitat multiplicated faster than those from the shady habitat, and the medium light intensity was optimal for the growth rate of both duckweed species. *Spirodela polyrhiza* showed high plasticity in terms of frond size, which increased under reduced light availability. *Lemna minor* from the shady habitat enhanced light harvest by increasing chlorophyll *a* concentration, but did not invest more in frond enlargement than *L. minor* from the open habitat. The inverse relationship between the rate of multiplication and frond enlargement suggests a trade-off between size and number in both duckweed species. Our results contribute to the knowledge about factors affecting duckweed communities and suggest that under shaded conditions *S. polyrhiza* is likely to achieve an advantage over *L. minor* due to the larger frond size of the former.

## ACKNOWLEDGEMENTS

We are grateful to Elżbieta Biardzka for her help in the chemistry laboratory and to Dr. Józef Klocek for making the growth room of the Institute of Biological Studies available .

### Funding

Funding was provided by the Ministry of Education and Science (Poland) through Siedlce University of Natural Sciences and Humanities grants nos. 386/14/S and 77/20/B. The funders had no role in study design, data collection and analysis, decision to publish, or preparation of the manuscript.

### Grant Disclosures

The following grant information was disclosed by the authors:
The Ministry of Education and Science (Poland) through Siedlce University of Natural Sciences and Humanities: 386/14/S, 77/20/B.

### Competing Interests

The authors declare there are no competing interests.

### Author Contributions

- Małgorzata Strzałek conceived and designed the experiments, performed the experiments, analyzed the data, prepared figures and/or tables, authored or reviewed drafts of the paper, and approved the final draft.
- Lech Kufel conceived and designed the experiments, performed the experiments, analyzed the data, authored or reviewed drafts of the paper, and approved the final draft.

### Data Availability

The raw data are available in the Supplementary File.

### Supplemental Information

Supplemental information for this article can be found online at http://dx.doi.org/10.7717/peerj.12698#supplemental-information.

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
