# Peer review of "Light intensity drives different growth strategies in two duckweed species: Lemna minor L. and Spirodela polyrhiza (L.) Schleiden"

_PeerJ, doi:10.7717/peerj.12698_

## Round 0.1 · original submission · Major Revisions

Noble Authors,

Your work was assessed by 3 experts. Everyone agreed that the work could be published. However, major corrections should be made before publishing. Please, read the comments of the reviewers and respond to them.

Best regards,

·

Basic reporting

The project based on the idea to compare isolated duckweed species from natural habitat with high or low light exposures. The authors selected Lemna minor and Spirodela polyrhiza and tested all isolates under three different light intensities under laboratory conditions. They tested both the increase of the number of fronds as well as the total frond area. This makes it possible to differentiate growth based on the number of fronds and growth based on increase of frond size and to distinguish different growth strategies. The experiments were planned well, the data sound very reliable (straight forward statistics) and the conclusions were well covered by the experimental data. Nevertheless, I am suggesting major revision by the following reasons.
Major critical points:
1. In Figures and Tables the experimental conditions are not well described.
Fig. 1: Which light conditions were used? Do not extend the curves beyond 16 d.
Fig. 2: Instead of “light variant”, the light intensity should be given. Which data are from which habitat, i.e. shaded and open? I refer in the next point to the calculation of growth rates.
Fig. 3: Which experimental light conditions were used?
Fig. 4: Please, repeat in the legend the calculation of S. The reader must be able to understand the figures without reading the complete text.
Fig. 5: Which experimental light conditions were used?
In general: the legends should describe everything to understand the figure. Similar problems might exist with the tables.
2. Material and Methods
I suggest to give the full Botanical names only here: Spirodela polyrhiza (L.) Schleid. and Lemna minor L. After the full name has been mentioned (perhaps without authority), the names should be given as S. polyrhiza and L. minor but not by mentioning just the genera.
The capital “M” means mol L-1. In light intensity, the correct dimension is µmol m-2 s-1.
The nutrient medium cited is now called “N medium” in literature. I suggest either using the term or giving the complete composition.
In the formulas, the authors used “log” instead of “ln”. I suggest urgently to change this because it will confuse any reader comparing the growth rates here and in the literature. Moreover, I suggest also urgently to call both rN and rS “relative growth rates”, perhaps qualified by the additional information “based on frond numbers” or “based on frond size”. The abbreviations could be used in the present form. This paragraph could be easier read when the equations are given in separate lines and perhaps numbered. This should be done with rN, rS, delta-r and Send/ Sstart.
The formula to calculate chlorophyll content should be given here.
3. Results
Whereas the statistical evaluation is very sophisticated, I miss the raw data of the results. This might be partially caused by the fact that it is not clear whet is presented in the figures. Perhaps the raw data were given in some of the figures. $Extending the legend might help.
4. Language
I am sorry to give my impression that the language is not always acceptable. I understood never the less everything. I feel obliged to give some examples:
- Line 22: “due to different resource use strategies than those of small duckweeds”
- Line 23: In
- 23 “In our study, the resource was light that influenced the plants before and during the laboratory experiment.“
- Line 48: “by turns”

5. Minor remarks
- Line 39/40: Lemna minor is not globally distributed – in contrast to S. polyrhiza. Both are finally not native in Europa.
- Line 42: What is meant by fertile? The fertility of the plants does not have an influence on forming mats.
- Line 47: The pockets in Wolffiella and Wolffia are not at the base of the fronds.
- Line 206: Perhaps it is better giving the chlorophyll content in mg
- Line 238-248: The lower light intensity might be easily caused by the fact that continuous white light is often used under laboratory conditions. I would hesitate to call the conditions used in the present project more natural. The bacteria associated with the different duckweed samples from the different places might have a dramatic influence on performance of the plants.
Line 281-283: Either more details should be provided or cancelled.
- The authors might like to know of a very new review paper covering the whole plant family: Acosta et al. (2021) Return of the Lemnaceae: duckweed as a model plant system in the genomics and postgenomics era. The Plant Cell doi:10.1093/plcell/koab189

Experimental design

See above

Validity of the findings

This is a quite interesting paper for a broad audience

Additional comments

The results are very interesting, the interpretaion is valid - I hope that authors are ready to improve their manuscript follwoing my suggestions.

·

Basic reporting

1. The manuscript is clearly written in professional, unambiguous language. If there is a weakness, it is in the statistical analysis (as I have noted above) which should be
improved upon before Acceptance.

Experimental design

2. What nutrient medium L114 (Hoagland’s or SH or other nutrient medium ) should be list in the manuscript, and what is the concentration of nutrient medium?
3. Lin 123: the experiment was carried out in thermostatic growth room at a temperature of 22oC and lasted 16 days. The difference temperature between day and night can also affect the growth of plants. Why difference temperature between day and night not set in the experimental design?

Validity of the findings

4. Is there turion produced during the S. polyrhiza cultivation process?
5. The English language should be improved to ensure that an international audience can clearly understand your text. Some examples where the language could be improved.
6. The following references can be cited in the manuscript to support your opinion.
The influence of different plant hormones on biomass and starch accumulation of duckweed: A renewable feedstock for bioethanol production[J]. Renewable Energy, 2019, 138(AUG.):659-665.
The effects of photoperiod and nutrition on duckweed (Landoltia punctata) growth and starch accumulation[J]. INDUSTRIAL CROPS AND PRODUCTS, 2018.

Additional comments

7. The workload of the article does not seem to be enough for a research manuscript, Protein content, starch content, and Chlorophyll a b and total should be also investigate.

·

Basic reporting

The manuscript is not well written, the sentences result often unclear because of their odd structure and there are several grammar mistakes. I recommend the authors to have the text reviewed by a person more familiar with English language. This is a friendly suggestion, please do not take this comment the wrong way. I am not an English native specker and I’m not stranger to grammar mistakes. I find very helpful to have somebody more proficient than me to proof-read my manuscripts before submission. It saves a lot of timey and gives the paper a higher chance to be accepted.

The results are not completely clear to me, probably just because of a written communication problem.

Experimental design

The work is original and the research question is clear. However, the importance of the study is not stated. Several experimental flows identified and listed below:

Lines 106-108 - The water surface in the pond is shaded by surrounding trees (shady habitat), while that in the bay is open (open habitat)

Are these evergreen species? Is the pond in low light conditions all the year or just in summer? This is very important. If in spring the pond is at different light conditions, a more opportunistic species could colonise it before shade-tolerant species. The ability to perform well in low-light conditions doesn’t necessarily represent an advantage if the pond is already taken by another species before the leaves of the trees restart their growth. Hence the shadow in summer is not necessarily what drives the compositions of the pond plant communities.

Line 110 Lemna minor and S. polyrhiza co-existed in both sites.

And which one was dominant? Was L. minor dominant in the shadow pond and S. polyrhiza dominant in the open pond? If yes, how did you assess the dominance?

Lines 114 – 115 filled with nutrient medium (Appenroth, Teller & Horn, 1996) and left for four-day-long preculture

Why the authors did not acclimatise the plants to the experimental conditions? Acclimatation is a fundamental step to be sure that the observations are due to the factors tested (in this case light intensity) and not to a stress reaction due to the sudden change of environmental conditions.

Lines 113- 115 Bulk plant material was brought to the laboratory, washed with tap water, placed in containers filled with nutrient medium (Appenroth, Teller & Horn, 1996) and left for four-day-long preculture

This is not clear, did the authors mix the fronds collected from the shadow and from the exposed environment? From lines 119-120 (In total, the number of cups was: 10 repetitions x 2 species x 120 2 sites x 3 light variants = 120.) I understand that both species, collected in both locations, were tested at all the three light intensities. If this is what you did, I agree with your methodology, but I do not see all these treatments showed in the results.

Lines 123 - 124The experiment was carried out in thermostatic growth room at a temperature of 22oC

What was the temperature of the ponds where the plants were collected? Since the plants were not acclimated to the experimental conditions, this information is very important. The two sites are very likely to differ in water temperature. Without acclimation you cannot exclude that the observations are due to the shock for sudden change in temperature.

Line 121 PPFD: 1 – 125, 2 – 236 and 3 – 459 µmol photons m–2 s–1 .

Why the authors did not test a very high light intensity, like the one measured in the exposed waterbody? The highest light intensity tested is still not really high and the tested lights are not very far away from each other.

Lines 131-134.

I see from the supplementary file that the biomass was also weighted. Why it is not shown in the results and why it was not used to calculate other interesting parameters? Light affects the thickness of the fronds. The total area of the fronds can’t, alone, be used to assess a strategy. Some species could reduce the fronds size, but produce thicker fronds. A complete morphological analyses should include Leaf Area Ratio (LAR). Net Assimilation Rate (NAR) also helps to understand the species strategy. In my opinion, the area and number of fronds alone are not sufficient to draw conclusions

Validity of the findings

I am not convinced of the validity of the findings as the plants, collected in different environments, were not acclimated to the experimental conditions.

Additional comments

Results
Lines 170 -171 Spirodela fronds grew and multiplicated slower at high light intensity

What Spirodela? The one collected from the shade of from the open habitat?

Lines 163 – 164 Univariate F tests 164 revealed that both species grew and reproduced fastest at medium light intensity

Is this valid for both plants collected from the shade and from the open habitat? Please specify.

Discussion
Lines 244 – 245 We showed that light conditions in the habitat of species origin affected the reproduction rate and thereby our findings confirm that the rate reflects the adaptation of clones to local conditions

How can the authors be sure that what they observed is adaptation and not acclimation?

Figure 1 Increasing number of fronds during experiment fitted to the exponential model. Data represent plants from the shady habitat grown at low light intensity (n = 10).

Please show also the results for plants collected in the exposed habitat

Figure 2 The growth and reproduction rates of Lemna minor and Spirodela polyrhiza from both habitats at different light intensities.

I don’t understand, from this caption I get that I should see for graphs: Growth rate and rate of reproduction for plants collected from low light environment and for plants collected from exposed environment.

Why only two graphs are shown? And what plants are these?

Figure 3, 4 and 5

Same comment as above, please show all the results. It’s not clear whether there is a difference between plants collected from low light environment and for plants collected from exposed environment.

Supplementary table is not very clear, please label better the columns. Also, why there are 4 light intensities (0, 1, 2, and 3)?

---

## Round 0.2 · Major Revisions

The reviewers carefully read the corrections that you brought to your work, following their suggestions. Two reviewers were of the opinion that the work could be published. However, the 3rd reviewer is still in doubt. I kindly ask you to respond to these remarks.

·

Basic reporting

The authors considered very carefully my suggestions and responded in a proper way. That they preferred to give the full Botanical name not in M&M but at the place of firts metioning is a matter of tast. I do not have a problem with it. I read also the response to teh other two reviewers and after that I suggest to accept the manuscript in the resent form.

Experimental design

Reveiwed in the original version - no problem

Validity of the findings

Reviewe din the original version - no problem

Additional comments

I suggest accepting the manuscript in the present form

·

Basic reporting

The language is strongly improved in this new version. Comparisons with studies that examined effect of light on these two species are strongly recommended. The Landolt book (Landolt, 1986. Biosystematic Investigations in the Family of Duckweeds (Lemnaceae), Vol. 2. Veroffent. Geobot. Inst. Eidg. Hochschule, Stift, Riubel, Zurich.) should have a section about effect of light on duckweed species.
The non-significant results were not included in the supplementary material, the authors only shared the results of the statistical tests. I am not convinced that the results of the study support the conclusion that Spirodela is better adapted to shady conditions.

Experimental design

As I highlighted before, the study is original and the research question is interesting and well defined. However there are two weaknesses in the experimental design: 1) lack of acclimation of the plants to the experimental conditions 2) Narrow range of light intensities tested. In particular, the authors did not test the performance of these species at high light intensitie (more than 800 micromol/m/sec).

Validity of the findings

Because of the weaknesses in the experimental design reported above, I am not convinced about the validity of the findings. I believe that if the authors are not willing to repeat the experiment they should change the conclusions and aknowledge clearly the limitations of the study. The authors stated in the respons to my comment that they agree with my doubts:
''We agree with the Reviewer’s doubts as to the possibility of far-reaching conclusions. The text has been modified in the line 259.'' But they didn't really address the matter in the discussion.

Additional comments

Answers to authors' answers

'The pond is surrounded by mixed, not evergreen forest (oak, maple, lime, pine trees). The first frond of duckweeds appear there at the same time as leaves on trees (the end of May). Therefore, shading is a factor operating for almost the whole growing season.'

This is a statement very difficult to prove. Moreover, I don’t see changes in your text that address this matter.

'there was no difference in the proportion of surface area occupied by the two species between open and shady habitats.'

Doesn’t this disprove your conclusions ‘Our results […]suggest that under shaded conditions, S. polyrhiza is likely to achieve an advantage over L. minor’ ??

'We acclimatised plants for 4 days at medium light intensity. Longer preculture periods would give us subsequent generations of fronds increasingly distant from natural conditions.'

I don’t understand, so you are testing acclimation, not adaptation? if you conclude that Spirodela performs better in shady environments because of its growth strategy, you are implying that it’s a genetic trait that determines the better performance at low light. If you don’t pre-adapt the plants to the experimental conditions, you can’t say that the better performance is due to its strategy, you can only conclude that plants acclimated (not adapted! Adaptation means that natural selection played a role in it and determined genetic traits that give the plant the ability to perform better in certain conditions) to shady conditions perform better in shady conditions. I strongly suggest repeating the experiment with plants acclimated to the experimental conditions.

'Light intensity measured in the open habitat was the highest possible on a sunny day. On cloudy and/or rainy days, the intensity of light was closer to our high experimental treatment.'

This doesn’t really convince me. Both sunny and cloudy days are present in real environmental conditions, the range of light intensities tested is not representative of the real one, you didn’t test high light conditions, so you don’t know how the two species behave when high intensities are present. This is a limitation that must be acknowledged in the manuscript.'

We agree with the Reviewer’s doubts as to the possibility of far-reaching conclusions. The text has been modified in the line 259.

I only see this: (however, see caveats raised by Ziegler et al., 2015). Which, in my opinion, is not enough to acknowledge the limitations due to lack of acclimation to experimental conditions.

M&M
Please add references for the equations used (rN = (lnNt – lnN0)/t, rS = (lnSt – lnS0)/16).

Discussion

Lines 259-260 this rate reflects the adaptation of clones to local conditions (however, see caveats raised by Ziegler et al., 2015).'
I disagree with the use of the term 'adaptation'. Also, here the authors should discuss properly the limitation of their study.

Conclusions

Lines 305-306 Lemna minor from the shady habitat enhanced light harvest by increasing chlorophyll a
concentration, but did not invest more in frond enlargement than L. minor from the open habitat.

The second species should be Spirorela

Lines 309-310 [...] suggest that under shaded conditions S. polyrhiza is likely to achieve an advantage over L. minor.

This is in contrast with your statement that Spirodela was not dominant in the shady site.

---

## Round 0.3 · accepted · Accept

Noble Authors,

I am pleased to inform you that your work meets all the requirements and can be published in its current version in PeerJ.
With best regards,